# Apigenin Targets MicroRNA-155, Enhances SHIP-1 Expression, and Augments Anti-Tumor Responses in Pancreatic Cancer

**DOI:** 10.3390/cancers14153613

**Published:** 2022-07-25

**Authors:** Kazim Husain, Krystal Villalobos-Ayala, Valentina Laverde, Oscar A. Vazquez, Bradley Miller, Samra Kazim, George Blanck, Margaret L. Hibbs, Gerald Krystal, Isra Elhussin, Joakin Mori, Clayton Yates, Tomar Ghansah

**Affiliations:** 1Department of Molecular Medicine, Morsani College of Medicine, University of South Florida, Tampa, FL 33612, USA; husaink@usf.edu (K.H.); krystalvilla@usf.edu (K.V.-A.); vlaverde@usf.edu (V.L.); oscarvazquez@usf.edu (O.A.V.); james121@usf.edu (B.M.); skazim1@mail.usf.edu (S.K.); gblanck@usf.edu (G.B.); 2Department of Immunology, H. Lee Moffitt Cancer Center, Tampa, FL 33612, USA; 3Department of Immunology and Pathology, Central Clinical School, Monash University, Melbourne 3004, Australia; margaret.hibbs@monash.edu; 4The Terry Fox Laboratory, BC Cancer, Vancouver, BC V5Z 1L3, Canada; gkrystal@bccrc.ca; 5Department of Biology and Center for Cancer Research, Tuskegee University, Tuskegee, AL 36088, USA; ielhussin0014@tuskegee.edu (I.E.); jmori0396@tuskegee.edu (J.M.); cyates@tuskegee.edu (C.Y.)

**Keywords:** SHIP-1, miR-155, tumor microenvironment, myelopoiesis, MDSC, TAM, pancreatic cancer, apigenin

## Abstract

**Simple Summary:**

Pancreatic Cancer (PC) is one of the most lethal cancers. PC promotes the expansion of immunosuppressive Myeloid-Derived Suppressor Cells (MDSC) and Tumor Associated Macrophages (TAM) that dampen anti-tumor immunity and renders immunotherapies ineffective. We have identified Src Homology-2 (SH2) domain-containing Inositol 5′-Phosphatase-1 (SHIP-1) as a potential new molecular target that controls myeloid associated immunosuppression. We previously demonstrated that apigenin (API) (bioflavonoid) increases SHIP-1 expression and promotes the development of monocytic-MDSC (M-MDSC) into M1 TAM (Tumoricidal) in the pancreatic tumor microenvironment (TME), which corresponded with an increase in anti-tumor immunity (tumor regression) mice harboring PC. In the present study, we demonstrate that API suppresses PC-induced micro-RNA (miR-155), a negative transcriptional regulator of SHIP-1 expression, something which corresponds with the augmented SHIP-1 expression that itself corresponds with an increase in the proportion of M1 macrophages in the bone marrow (BM) of mice with PC. This work supports the idea of enhancing SHIP-1 as a potential new therapeutic target for treatment of pancreatic cancer.

**Abstract:**

Pancreatic cancer (PC) is a deadly disease with a grim prognosis. Pancreatic tumor derived factors (TDF) contribute to the induction of an immunosuppressive tumor microenvironment (TME) that impedes the effectiveness of immunotherapy. PC-induced microRNA-155 (miRNA-155) represses expression of Src homology 2 (SH2) domain-containing Inositol 5′-phosphatase-1 (SHIP-1), a regulator of myeloid cell development and function, thus impacting anti-tumor immunity. We recently reported that the bioflavonoid apigenin (API) increased SHIP-1 expression which correlated with the expansion of tumoricidal macrophages (TAM) and improved anti-tumor immune responses in the TME of mice with PC. We now show that API transcriptionally regulates SHIP-1 expression via the suppression of miRNA-155, impacting anti-tumor immune responses in the bone marrow (BM) and TME of mice with PC. We discovered that API reduced miRNA-155 in the PC milieu, which induced SHIP-1 expression. This promoted the restoration of myelopoiesis and increased anti-tumor immune responses in the TME of heterotopic, orthotopic and transgenic SHIP-1 knockout preclinical mouse models of PC. Our results suggest that manipulating SHIP-1 through miR-155 may assist in augmenting anti-tumor immune responses and aid in the therapeutic intervention of PC.

## 1. Introduction

Pancreatic cancer (PC) is an aggressive and lethal malignancy. Based on its increasing incidence it is projected to become the second leading cause of cancer-related death after lung cancer in the US by 2030 [1,2,3,4]. The lethality is attributed to late diagnosis, early metastasis, and limited response to current chemotherapies: gemcitabine, nab-paclitaxel and FOLFIRINOX [5]. Despite advances in immunotherapy for the treatment of multiple solid tumors, the role of immunotherapy in PC remains limited because inflammatory tumor-derived factors (TDF) contribute to the induction of an immunosuppressive tumor microenvironment (TME). This immunosuppressive TME inactivates anti-tumor immune responses by halting the recruitment of effector immune cells into the tumor [6,7,8]. The inflammatory TME does this, in part, by inducing the expansion of regulatory immunosuppressive myeloid-derived suppressor cells (MDSC) generated from the bone marrow (BM) and found in human PC and pre-clinical PC models. These MDSC consist of immature myeloid cells, macrophages, granulocytes and dendritic cells (DC) that aggressively suppress anti-tumor immunity via multiple modalities [6,7,8,9]. Furthermore, expansion of MDSCs has also been reported in the BM and tumors of PC patients compared with healthy controls [10] and these MDSC have been reported, in pre-clinical models of PC, to be responsible for the suppression of anti-tumor immune responses [11,12]. In addition, granulocytic and monocytic MDSC (M-MDSC), mobilized into the TME and M-MDSC, can differentiate into protumor M2 tumor-associated macrophages (M2-TAM) vs. tumoricidal M1 TAM, which have the potential to further suppress anti-tumor immune responses, and promote metastasis and chemoresistance in PC [13,14,15,16].

MicroRNAs (miRs) are small, nonprotein-coding RNAs (18–24 nucleotides in length) that can inhibit gene expression at the post-transcriptional level through binding to the complementary sequences of their target mRNAs at 3′-untranslated regions (3′-UTRs) [17]. They have been shown to regulate biological processes such as cell proliferation, cellular differentiation, stem cell development, homeostasis and apoptosis, consequently affecting biological events such as cell survival, immune modulation and carcinogenesis [17,18,19,20]. Deregulation of miRNAs has been associated with almost all human malignancies, either acting as oncogenes (OncoMirs) or tumor suppressors [21]. One of the first miRNAs identified with oncogenic potential was miR-155 which was found to be overexpressed in lymphoma, breast, colon and PC [21,22]. MiR-155 presented the highest prognostic impact in PC patients linked to poor survival [23,24,25]. MiR-155 targets Src Homology-2 (SH2) domain-containing Inositol 5′-Phosphatase-1 (SHIP-1) transcription and thus regulates inflammation, MDSC activation and polarization of TAM [26,27,28].

SHIP-1 is a 145 kDa protein that regulates the activity of macrophages [29,30,31,32]. SHIP-1 expression is regulated in immune cells by external soluble factors such as cytokines and chemokines in the microenvironment [33]. We, and others, have shown that SHIP-1 knockout (KO) mice have markedly increased numbers of immunosuppressive, pro-tumor M2 macrophages, demonstrating the role of SHIP-1 in regulating macrophage polarization [12,29,34]. In addition, we have reported that downregulation of SHIP-1 protein expression and expansion of MDSC corresponds with an increase in tumor burden in mice with PC [11,35]. Therefore, we proposed that suppression of miR-155 production might promote upregulation of SHIP-1 expression and thus restore M-MDSC homeostasis, increase tumoricidal M1-TAM percentages and improve anti-tumor immune responses in PC.

Several studies have suggested that increased intake of dietary fruits, vegetables, and cereal grains may prevent gastrointestinal cancers, including PC [36,37,38]. Recently, natural compounds known as bioflavonoids including apigenin (API) have been demonstrated in both in vitro and in vivo models to exert broad anticancer activities in a variety of malignancies such as breast cancer [39], liver cancer [40], prostate cancer [41], lung cancer [42], colon cancer [43], melanoma [44], osteosarcoma [45] and PC [12,46,47]. API inhibits tumor cell proliferation by inducing apoptosis leading to autophagy and cell cycle arrest at the G_2_/M phase and can also reduce cancer cell motility, thereby preventing cancer cell migration and invasion regulating PI3K/AKT, MAPK/ERK, JAK/STAT, NF-κB, p53 and Wnt/β-catenin signaling pathways [48,49]. API has demonstrated potent anti-tumor activity and the ability to reduce chemoresistance to gemcitabine (one of the chemotherapy drugs used for PC) in human PC cell lines [50]. API has also been shown to induce apoptosis through p53-dependent and p53-independent mechanisms [51]. The induction of apoptosis induced by API involves both extrinsic and intrinsic pathways in cancer cells [52,53]. Apoptosis targets of API consists of caspase-3, -8, and -9, Bax, Bak, Bad, Bim, Bid, Bcl-xL, XIAP, Mcl-1, Bcl-2, m-TOR/PI3K/AKT, STAT3, p53, p21, p27, PARP cleavage, FOXO3a, AIF, Apaf-1, DR5, ERK/JNK/p38 MAPK, Jun, NF-κB, Noxa, PUMA, Smac, Survivin, FAS and TRAIL [51]. API has shown the most selective killing of cancer cells while sparing normal cells [54]. Our research group recently reported that API reduced tumor burden, improved anti-tumor immune responses and increased survival rates of mice bearing pancreatic tumors compared with vehicle treated mice with PC [12,55]. However, the ability of API to target miR-155 and SHIP-1 expression in PC has not been studied. Therefore, in this study we have explored the molecular mechanism by which API acts. We have found that it targets miR-155, enhancing SHIP-1 expression, which leads to a restoration of MDSC homeostasis, and an increase in tumoricidal M1-TAM percentages thereby improving anti-tumor immune responses in mice with PC.

## 2. Materials and Methods

### 2.1. Pancreatic Ductal Adenocarcinoma Cancer Cell Lines

Human PC cell line MiaPaCa-2 was obtained from Mokenge Malafa of H. Lee Moffitt Cancer Center, USA. The murine Panc02 adenocarcinoma cell line originated from C57BL/6 mice [56]. The murine UN-KC-6141 cell line was derived from a C57BL/6 mouse bearing a Kras^G12D^; Pdx1-Cre (KC) pancreatic tumor [57]. Panc02, MiaPaCa-2 and UN-KC-6141 cell lines were maintained in RPMI 1640 or DMEM (+4.5 g/L D-Glucose, L-Glutamine), supplemented with 10% fetal bovine serum (FBS) (HyClone), 100 U/mL penicillin and 100 μg/mL streptomycin (Gibco, Waltham, MA, USA) at 37 °C in 5% CO_2_ incubator. Cultured cells were negative for mycoplasma and viral contamination.

### 2.2. Bioflavonoid

Apigenin (API) (4′,5,7-Trihydroxyflavone, 5,7-Dihydroxy-2-(4-hydroxyphenyl)-4-benzopyrone) (Sigma-Aldrich, St. Louis, MO, USA) was diluted in DMSO (100 mM), stored at −20 °C and later used for assays described in this study.

### 2.3. Cell Viability and Apoptosis Assays of Pancreatic Cancer Cells

The methyl thiazol tetrazolium (MTT) assay was performed as described previously [58]. Briefly, MiaPaCa-2, Panc02 and UN-KC-6141 cells were seeded in 96-well plates at a density of 3000 cells/well and allowed to attach overnight. PC cells were then treated with API (40 µM) or DMSO (1%) as vehicle control for 24 h at 37 °C in 5% CO_2_ incubator. After 24 h, media was replaced with 20 µL of MTT (1 mg/mL) (Sigma-Aldrich) and incubated for 3 h at 37 °C in 5% CO_2_. MTT was replaced with DMSO, and absorbance was read at 570 nm. 

Panc02 and UN-KC-6141 cells were seeded in 6-well plates allowed to grow to 50–60% confluency. Cells were then treated with API at a dose-dependent concentration (10, 20, 30, 40, 50 µM) or DMSO (1%) as vehicle control and incubated for 24 h at 37 °C in a humidified atmosphere of 5% CO_2_. Cells were collected and stained with a FITC Annexin V Apoptosis Detection Kit I (BD Bioscience, San Jose, CA, USA) according to the manufacturer’s instructions. Acquisition of samples was performed using a flow cytometer BD LSRII (BD Biosciences Immunocytometry Systems, San Jose, CA, USA). FlowJo v10.8 so ftware (TreeStar Inc., Ashland, OR, USA) was used for data analysis.

### 2.4. Transfection of miR-155 Inhibitor into Pancreatic Cancer Cells

MiaPaCa-2, Panc02 and UN-KC-6141 cells (2 × 10^5^) were grown in 2 mL of growth medium in a 6-well tissue culture plate for 18–24 h until 60–80% confluency. The following solutions were prepared: Solution A) For each transfection, diluted 1 µL of miRNA-155 inhibitors (1 µM) (Table 1) or scrambled negative control miRNA (1 µM) (Qiagen, Germantown, MD, USA) into 50 µL Opti-MEM Transfection Medium (Invitrogen, Waltham, MA, USA). Solution B) For each transfection, diluted 3 µL of Hi-Perfect Transfection Reagent into 50 µL Opti-MEM Transfection Medium. Solution A was added directly to Solution B, mixed gently and the mixture incubated for 10–15 min at room temperature. Cells were then washed once with 2 mL of Opti-MEM Transfection Medium and 900 µL of Opti-MEM Transfection Medium was added to each tube containing 100 µL (Solution A + Solution B), mixed gently and the mixture was overlaid onto the washed cells drop wise. The cells were incubated for 24 h at 37 °C in a CO_2_ incubator. After 24 h post-transfection with miRNA-155 inhibitors or scrambled negative control miRNA, cohorts of cells were treated with either API (40 µM) or DMSO (1%) as vehicle control for 24 h at 37 °C in 5% CO_2_ incubator. Cells were harvested for different biological assays as described within the cell viability and miRNA extraction sections.

### 2.5. Pancreatic Cancer Murine Models

All female C57BL/6 mice (6–8 weeks of age) described in this study were purchased from Envigo (Indianapolis, IN, USA) and were acclimatize for one week in a pathogen-free animal facility (University of South Florida (USF) vivarium) before injections with PC cells.

To generate heterotopic pancreatic cancer (HPC) models, mice were subcutaneously injected with 1.5 × 10^5^ murine Panc02 (HPC mice) or 5 × 10^6^ murine UN-KC-6141 cells (KC-HPC mice), in sterile 1x phosphate buffer saline (PBS), in the lower ventral abdomen, while control (CTRL) mice received sterile 1x PBS. Once tumors where palpable, API (25 mg/kg, 100 µL volume) treatments (HPC-API and KC-HPC-API), via intraperitoneal (IP) injections, were started while CTRL, HPC and KC-HPC mice received sterile PBS (vehicle) three times per week until the end of the study [12,55]. The HPC models were euthanized 21–28 days post-injection, while the KC-HPC models were euthanized 16–17 days post-injection.

To generate orthotopic pancreatic models (OPC), anesthetized (1.5–3% isoflurane) mice were injected in the neck of the pancreas, via laparotomy [59], with sterile PBS (CTRL) or 1.25 × 10^4^ Panc02 cells (OPC), in sterile 1x PBS. Once pancreatic tumors were confirmed through ultrasound imaging (Vevo 2100), a cohort of OPC mice received API (25 mg/kg, 100 µL volume) while CTRL and OPC mice received sterile PBS (vehicle), all via IP, three times per week until the end of the study [12]. The endpoint of OPC mice was reached 16–20 days post-surgery.

Male and Female SHIP^HET^ mice (C57BL/6 background) were kindly received from the Hibbs Lab [60]. These mice were bred at the USF vivarium to obtain SHIP^KO^ and SHIP^WT^, confirmed by genotyping. To generate SHIP^KO^-HPC and SHIP^WT^-HPC mice, these mice (4–6 weeks of age) were SC injected with 1.5 × 10^5^ Panc02 cells as previously described above. The endpoint of the study was reached 14 days post-injection.

All our PC models along with their CTRL counterparts were humanely euthanized (CO_2_ and cervical dislocation), according to the USF Institutional Animal Care and Use Committee (IACUC) guidelines, as described in our previous study [12]. The approved IACUC protocols IS00004664 and IS00004665 abides by the Guide for the Care and Use of Laboratory Animals. Mouse tissues including, tumors and femur/tibia, were harvested and processed for different biological assays as described in further sections of this study.

### 2.6. Flow Cytometry

The femur and tibia were cut at the pelvic-hip joint and muscle was removed. The femurs and tibias were cut at the ends and flushed with a 30-gauge needle and 10 cc syringe filled with completed RPMI media until bone was white onto a petri dish. Pelleted bone marrow (BM) cells were treated with Red Blood Cell (RBC) lysis buffer (eBioscience, Waltham, MA, USA), according to the manufacturer’s instructions, and then filtered through a 70 μM nylon mesh cell strainer in 1x PBS. All pancreatic tumors were manually cut into ~1–2 mm^3^ fragments with a sterile razor blade and then enzymatically digested with Collagenase, type IV (Sigma-Aldrich), DNase, type IV (Sigma-Aldrich) and Hyaluronidase, Type V (Sigma-Aldrich) in Hank’s Balanced Salt Solution for 1 h, spun down and washed as described in previous studies [61]. Digested pancreatic tumor samples were treated with RBC lysis buffer and then filtered through a 70 μM nylon mesh cell strainer in 1x PBS. Bone marrow and digested pancreatic tumor single cell suspensions were counted with a hemocytometer and then resuspended in 3% FBS/PBS. BM and digested pancreatic tumor cells (1 × 10^6^) were Fc Blocked (anti-mouse CD16/CD32) and then surface stained for 30 min on ice protected from light with fluorescent anti-mouse antibodies including, CD11b-APC, Ly6C-PE/Cy7, Ly6G-PerCP, F4/80-BV650, MHC-II-BV605, CD206-FITC and CD40-PE/Cy5 along with isotype control antibodies (Biolegend, San Diego, CA, USA) to detect MDSC and macrophage subsets. Digested pancreatic tumor cells (1 × 10^6^) were also Fc Blocked and then surface stained with fluorescent anti-mouse antibodies including CD3-FITC, CD8-PercP/Cy5.5, CD4-APC/Cy7, and CD40L-APC along with isotype control antibodies (Biolegend) to detect CD8^+^ T cells. Subsequently, all samples were fixed with 2% paraformaldehyde for 15 min on ice protected from light. Acquisition of samples was performed using a flow cytometer BD LSRII (BD Biosciences Immunocytometry Systems). Data analysis was performed using FlowJo v10.8 software (TreeStar Inc.).

### 2.7. miRNA Extraction and TaqMan miRNA Assay

miRNA extraction of PC cells (treated with miR-155 inhibitor and/or API), BM and digested pancreatic tumors cells was performed using a mirVana miRNA Isolation Kit (Applied Biosystems, Waltham, MA, USA) as per the manufacturer’s instructions. miRNA was reverse transcribed into cDNA using TaqMan miRNA reverse transcription kit (Applied Biosystems) with gene-specific stem-loop RT primers according to the manufacturer’s instructions. cDNA was then loaded onto either mmu-miR-155 or ipu-miR-155 TaqMan miRNA assays (Assay IDs: 002571 and 467534_mat) using TaqMan Fast Universal PCR Master mix (Applied Biosystems) and miR-155 detected with an Eppendorf Master cycler real plex 4. The PCR cycling parameters were 95 °C for 10 min followed by 40 cycles of a denaturing step at 95 °C for 15 s and an annealing/extension step at 60 °C for 1 min. All reactions were performed in triplicate. Relative expression of miR-155 was calculated using the comparative 2^−∆∆Ct^ method normalized to a mouse endogenous control gene snoRNA202 and human endogenous control gene U6 (Assay IDs: 001232 and 001093).

### 2.8. RT-Quantitative PCR (RT-qPCR)

Isolation of total RNA from digested pancreatic tumors and BM cells was performed using a RNeasy Mini Kit (Qiagen), according to the manufacturer’s instructions. RT-PCR of quantified and normalized total RNA was performed using a High-Capacity cDNA RT Kit with an RNase Inhibitor (Applied Biosystems), according to the manufacturer’s instructions. SHIP-1 and GAPDH (housekeeping gene) mRNA levels were evaluated with an Eppendorf Master cycler real plex 4 using iQ SYBR Green Supermix (Bio-Rad, Hercules, CA, USA) along with the following primers from Integrated DNA Technologies: SHIP-1 forward, 5′-CCA GGG CAA GAT GAG GGA GA-3′, SHIP-1 reverse, 5′-GGA CCT CGG TTG GCA ATG TA-3′, and GAPDH forward, 5′-TGA TGG CGT GGA CAG TGG TCA TAA-3′, GAPDH reverse, 5′-CAT GTT TGT GAT GGG CGT GAA CCA. qPCR of each sample was performed in triplicate and under the following conditions: 95 °C for 3 min followed by 40 cycles of 95 °C for 15 s and 60 °C for 1 min. The comparative 2^−∆∆Ct^ method was used to evaluate relative SHIP-1 mRNA levels of digested pancreatic tumors and BM cells.

### 2.9. Western Blot (WB)

Single-cell suspensions of BM, digested pancreatic tumors cells and UN-KC-6141 cells treated with API (40 µM) were lysed with radioimmunoprecipitation buffer (Sigma-Aldrich), quantified with a BCA Protein Assay kit (Thermo Fisher Scientific, Waltham, MA, USA), resolved by SDS-PAGE (Thermo Fisher Scientific) and transferred onto a nitrocellulose membrane as previously described [12]. Membranes were probed with either anti-SHIP-1 (Santa Cruz Biotechnology, Dallas, TX, USA), anti-iNOS (Cell Signaling Technology) or anti-Bcl-2 (Cell Signaling Technology, Danvers, MA, USA), all at a dilution of 1:1000. All blots were re-probed with a housekeeping protein, HRP-conjugated anti-β-actin (Sigma-Aldrich), at a dilution of 1:20,000. Membranes were then probed with the respective secondary anti-mouse or rabbit IgG HRP-conjugated antibodies (1:1000) of SHIP-1 and iNOS. Detection of SHIP-1, iNOS and β-actin proteins was performed using Super Signal West Pico or Femto Chemiluminescent Substrates (Thermo Fisher Scientific) and then imaged on a Bio-Rad Chemi Doc XRS Imaging System. Normalized densitometric ratios (divided by β-actin) were determined by quantifying WB results using Image J.

### 2.10. Akoya Codex

The poly-l-lysine coated coverslips containing 5-µm of Fresh Frozen (FF) tissue sections were stored at −80 °C until use. At the staining time, FF tissue sections were baked for 30 min to 1 h at 55 °C. According to Akoya Biosciences protocol, the FF tissues were dewaxed, deparaffinized in xylene then rehydrated in descending ethanol concentrations (100% twice, 90%, 70%, 50%, and 30%, respectively) and washed in ddH_2_O twice, each step for 5 min. Heat-induced epitope retrieval with antigen retrieval solution, pH 6, was performed using the pressure cooker at high-pressure protocol (80 °C) for 20 min. After cooling at room temperature (RT) for 30 min to 1 h, the coverslips were washed in ddH2O twice for 2 min. Then the sample coverslips were immersed in the hydration buffer six times before being placed in the staining buffer for 20–30 min. The anti-body cocktail solution was prepared to contain 1–2 µL: 200 of the antibody/sample and then added to CODEX blocking buffer (staining buffer, N blocker, G blocker, J blocker, and S blocker) to block the nonspecific binding of the antibody. For each coverslip, 190 µL of the antibody cocktail solution was added and incubated in a sealed humidity chamber for 3 h at RT or overnight at 4 °C. After staining, sample coverslips were placed in the staining buffer twice for 2 min to rinse any unbound antibodies and then fixed in 1.6% paraformaldehyde diluted in the storage buffer (post-staining fixing solution) for 10 min, followed by a total of 9 quick washes in 1x PBS. After washing, the sample coverslips were incubated in 100% cold methanol for 5 min, followed by a total of nine dunks in 1x PBS. A fresh final fixative solution was prepared by diluting 20 µL of the CODEX fixative reagent in 1 mL of 1x PBS. The final fixative solution (190 µL) was added to the sample and incubated in a sealed humidity chamber at RT for 20 min, followed by nine quick washes in 1x PBS to remove the fixative reagent. Thereafter, sample coverslips were placed in a storage buffer at 4 °C for up to two weeks or further processed for imaging. At imaging time, the reporters’ plate was prepared for the corresponding antibodies (one well/cycle), maintaining one dye type per cycle. The reporter stock solution was prepared for the total number of cycles. Each reporter was added (5 µL) to the corresponding cycle to create a reporter master mix per cycle, then gently mixed by pipetting before 245 µL of the mix was added into the corresponding well on the 96-well plate.

Images were collected using a KEYENCE BZ-X800 fluorescent microscope config-ured with 3 fluorescent channels (TxRed, Cy7, Cy5) and DAPI with 20×. Each tissue was imaged with a 20× oil immersion objective in a 5 × 5 tiled acquisition at 9 z-planes per tile. Images were subjected to deconvolution to remove out-of-focus light. Then the raw experiment data were transferred using CODEX Instrument Management version (CIM v1.29) software and processed using CODEX Processor version 1.7. The processed data were analyzed using the CODEX-MAV plugin in Image J. Then, nuclei segmentation was performed, followed by gating and clustering of the CD8^+^ T cells populations. Further analyses were performed using the Seurat package in R software (version 4.1) to generate dimensional reduction and clustering of the population of interest. The data were visualized using 2-dimensional plots (UMAP & tSNE plot).

### 2.11. The Cancer Genome Atlas (TGCA) Database

miR-155 and SHIP-1 RNAseq values from PC patients were obtained from cbioportal.org.

### 2.12. Statistical Analysis

Results are presented as mean ± Standard Deviation (S.D.) of all in vitro and in vivo experiments of at least three independent biological replicates. Significant differences were considered at *p* < 0.05 when analyzed by unpaired two-tailed *t* tests using Prism 8 Software (GraphPad, San Diego, CA, USA).

## 3. Results

### 3.1. API Suppresses miR-155 Gene Expression and Inhibits the Viability of Pancreatic Cancer Cells In Vitro

MiR-155 is known to be overexpressed in different cancers, including PC [21,23]. We first assessed whether apigenin could modulate the expression of miR-155 in PC cells. Murine PC cell lines Panc02 and UN-KC-6141 as well as the human PC cell line MiaPaCa-2 were treated with API (40 µM) or DMSO vehicle for 24 h and the expression of miR-155 was examined using RT-qPCR assay. We found that API significantly suppressed miR-155 gene expression in three PC cell lines (Figure 1). Since API is known to inhibit the growth of cancer cells, including PC cells [39,40,41,42,43,47,50], we used the same PC cell lines treated with API (40 µM) or DMSO (vehicle) for 24 h and examined cell viability using MTT assay. Our data show that API significantly inhibited the viability of PC cell lines (Figure 1), indicating the anti-carcinogenic effect of API in PC. Moreover, we then treated Panc02 and UN-KC-6141 cells with API at a dose dependent concentration (10, 20, 30, 40, 50 µM) and detected apoptosis using flow cytometry (Annexin V and PI). Our results show that API induces apoptosis in both Panc02 and UN-KC-6141 cells with increasing concentration of API (Appendix A). We then assessed Bcl-2 protein expression in UN-KC-6141 cells treated with API (40 μM) and our WB results show a significant decrease in Bcl-2 protein in API treated PC cells (Appendix A).

### 3.2. Inhibition of miR-155 Decreases the Viability of PC Cells In Vitro

miR-155 is implicated in the development of pancreatic tumors from pancreatic intraepithelial neoplasia (PanIN) lesions, and is overexpressed in PC [23]. We then asked whether the depletion of miR-155 modulated PC viability. Murine and human PC cell lines were transfected with miR-155 inhibitor (i.e., MMU-MIR-155-5P/HSA-MIR-155-5P) and scrambled miRNA at a concentration of 100 nM for 24 h and cell viability determined using an MTT assay. As well, the expression of miR-155 was examined using an RT-qPCR assay. Our results show that miRNA-155 inhibitor significantly depleted miR-155 gene expression, and correspondingly, suppressed the viability of three different PC cell lines compared to scrambled miRNA (Figure 2A–F), indicating the vital role that miR-155 plays in the growth of PC.

### 3.3. Concomitant API and miR-155 Inhibitor Treatment Synergistically Suppresses miR-155 and the Viability of PC Cells In Vitro

Since miR-155 is overexpressed and implicated in the development of PC [23], we then asked if combining a miR-155 inhibitor with API further inhibits PC cell viability. To test this, the murine PC cell line, UN-KC-6141, was transfected with either miR-155 inhibitor or scrambled miRNA and treated with API and cell viability and miR-155 gene expression were examined. Interestingly, the combination of API and miR-155 inhibitor synergistically depleted miR-155 gene expression in PC cells and suppressed PC cell viability (Figure 3A,B), suggesting the anti-cancer activity of both API and miR-155 inhibitor against PC.

### 3.4. API Decreases the Production of miR-155 and This Correlates with an Increase in SHIP-1 Expression in HPC, OPC and KC-HPC Mouse Models

Earlier studies have shown the anti-tumor activity of API in pre-clinical experimental models of cancer [41,62,63,64,65,66,67,68]. We have reported that API increased survival and reduced tumor growth in a heterotopic (H) mouse model of PC (HPC) [55]. To investigate if API targets miR-155 and regulates SHIP-1 in the TME of an HPC mouse model, HPC mice were treated with API at a dose of 25 mg/kg (IP) 3 times per week for 3–4 weeks. Our results show that API treatment significantly reduced miR-155 gene expression which, in turn, increased SHIP-1 gene and protein expressions in the tumor of HPC mice compared with vehicle treated HPC mice (Appendix A).

We reported recently that API reduced tumor growth in an orthotopic (O) mouse model of PC (OPC) [12], which is a more clinically relevant model [69]. To investigate if API targets miR-155 and regulates SHIP-1 expression in the BM and tumor of an OPC mouse model, OPC mice were treated with API at a dose of 25 mg/kg (IP) 3 times per week for 2–3 weeks. We observed a significant increase in miR-155 gene expression in the BM of OPC vs. CTRL mice, and found that API treatment of OPC mice inhibits the overexpression of miR-155 (Figure 4A). In contrast, SHIP-1 gene and protein expression in the BM was significantly increased in API-treated OPC mice compared with OPC mice (Figure 4B–D). Looking at the pancreatic tumors from OPC mice, we observed very marked overexpression of miR-155 compared with the pancreas of CTRL mice, however due to API treatment of OPC mice a significant decrease in miR-55 gene expression was noticed (Figure 4E). The finding of increased SHIP-1 expression in the BM of API-treated OPC mice suggests that it may be involved in the regulation of myelopoeisis. In addition, these results corroborate our previous findings that SHIP-1 gene and protein expressions is increased in the pancreatic tumor of API-treated OPC-API mice [12].

To build on these findings, we then used a novel pancreatic cancer cell line UN-KC-6141 that resembles human PC due to its expression of a mutated *Kras* and generated mice harboring these tumor cells, which we called KC-HPC mice. Such mice were treated with API at a dose of 25 mg/kg (IP) three times per week for 17 days. We investigated if API also targets miR-155 and regulates SHIP-1 in the tumors from KC-HPC mouse model. We found that API treatment significantly reduced miR-155 gene expression in the tumors from KC-HPC mice compared with vehicle treated KC-HPC mice (Figure 5A), and significantly increased the gene and protein expressions of SHIP-1 (Figure 5B–D).

### 3.5. miR-155 Expression in SHIP^KO^-HPC Is Increased Compared to SHIP^WT^-HPC Mice

SHIP-1 is known to be involved in the development and function of myeloid cells including MDSC, macrophages and DC [30] and is one of the targets of miR-155 [28,30], impacting tumor immunity. Earlier we reported downregulation of SHIP-1 expression in immunocompetent C57BL/6 mice with PC [11]. To investigate the role of miR-155 and SHIP-1 in PC, murine Panc02 cells were heterotopically inoculated into SHIP^WT^ mice and SHIP^KO^ mice, which we designated SHIP^WT^-HPC and SHIP^KO^-HPC mice, respectively. We observed significantly higher miR-155 expression in the BM and tumors from SHIP^KO^-HPC mice compared with SHIP^WT^-HPC mice (Figure 6A,B).

### 3.6. API Improves Myelopoiesis and Anti-Tumor Responses in KC-HPC Mice

Pancreatic cancer patients are known to have altered myelopoiesis and since API has been shown to target miR-155 in microglia and macrophages [10,70,71,72], we then examined MDSC and macrophage subsets in the BM of KC-HPC mice. Flow cytometric analysis of BM cells showed that KC-HPC mice had significantly higher percentages of M-MDSCs compared with control mice and API treatment significantly reduced proportions of G-MDSCs without any significant change in proportions of M-MDSCs compared with vehicle treated KC-HPC mice (Figure 7A,B). We calculated the absolute cell numbers of MDSC subsets in the BM of KC-HPC mice, we observed that both G-MDSC and M-MDSC cell numbers were significantly increased and API treatment significantly lowered MDSC subsets absolute cell numbers (Appendix A). Since macrophages in the BM can be phenotypically characterized as M1 or M2 macrophages, which are tumoricidal and pro-tumor, respectively [73], we carried out flow cytometric analysis of the BM of KC-HPC mice and found a significant decrease in M1 and increase in M2 macrophages compared with control mice (Figure 7C,D). Interestingly, API treatment of KC-HPC mice significantly increased M1 and decreased M2 macrophage percentages in the BM compared with KC-HPC mice (Figure 7C,D). We also observed that the absolute cell numbers of M1 macrophages were significantly decreased whereas M2 macrophages were significantly increased in BM of KC-HPC mice, however there were no significant changes in M1 and M2 macrophages absolute cell numbers of KC-HPC mice treated with API (Appendix A).

In keeping with our previous report showing that API decreases the tumor weights of PC mice [12,55], we found herein that API slows the growth of tumors in KC-HPC mice, significantly reducing tumor burden (Figure 8A–C). We then assessed the expression of CD40 (a T cell costimulation marker) on M1 TAM (gated as MHC-II^+^ and CD206^−^) within the tumor from KC-HPC vs. KC-HPC-API mice. Flow cytometry showed that CD40 expression was significantly upregulated on M1 TAM and this correlated with a significant increase in expression of iNOS protein in the tumor of KC-HPC-API compared with KC-HPC mice (Figure 9A–D). Flow cytometric analysis and calculated absolute cell numbers revealed a significant increase in the intra-tumoral CD8^+^ T cells (CD3^+^CD8^+^CD4^−^) infiltrating the tumor of API-treated KC-HPC mice compared with vehicle-treated KC-HPC mice, however there were no significant changes in their expression of CD40L (Figure 10A–E). Next, we examined the expression and infiltration of CD8^+^ T cells and MHC-II^+^ cells in the tumor sections of KC-HPC treated with vehicle or API and using AKOYA-CODEX which provides spatial capture of location and phenotype for multiple immune cells in the tumor. Two-dimensional UMAP plots of single cells show that API treated KC-HPC mice demonstrated an increase in MHC-II^+^ cells which was non-detectable in KC-HPC mice (Appendix A). We then observed, via 2D tSNE, increased CD8^+^ T cell expression partially located within the tumor plots for the tumors of KC-HPC-API mice compared with vehicle treated KC-HPC mice (Appendix A).

### 3.7. Lower Survival Is Associated with High miR-155 and Low SHIP-1 Gene Expressions in PC Patients

To determine if augmented miR-155 expression in PC patients correlated with reduced SHIP-1 expression, we obtained the RNAseq values for The Cancer Genome Atlas-pancreatic adenocarcinoma database (cbioportal.org) and assembled two groups of case IDs representing (i) high miR155HG expression and low SHIP-1 expression versus (ii) low miR155HG and high SHIP-1 expression (Appendix A). The overlaps were based on the upper and lower 50th percentiles for both groups. For example, group (i) represents the overlap of the upper 50th percentile of miR155HG expression and the lower 50th percentile of SHIP-1 expression (Appendix A). While the Kaplan–Meier analysis did not indicate a log-rank p-value representing a distinction between the two sets of case IDs, at the 10-month time period there was a significant difference in survival in PC patients, with lower survival correlating with high miR155HG and low SHIP-1 expression (Table 2).

## 4. Discussion

The efficacy of cancer immunotherapy in PC patients remains limited because of the induction of an immunosuppressive TME, which leads to the inactivation of anti-tumor immune responses [6,7,74]. Targeting a TME enriched with immunosuppressive cells, specifically MDSC and TAM, is an important strategy to improve the success of immunotherapy in PC. Pre-clinical studies have elucidated the critical role of MDSC and TAM not only in pancreatic tumor progression and metastasis but also in conferring resistance to chemotherapy [75,76]. Furthermore, accumulation of MDSC and pro-tumor TAM in the TME have been shown to correlate with metastatic relapse, leading to reduced survival in PC patients [77,78]. In our earlier preclinical studies, we reported that apigenin depleted immunosuppressive MDSC and TAM from the TME, induced SHIP-1 expression, increased tumoricidal macrophages, enhanced anti-tumor immune responses and reduced tumor burden in different PC models [12]. In the present study, we have found that apigenin depleted miR-155 expression in murine as well as human PC cell lines and this corresponded with our current results regarding increased apoptosis and reduced cell viability. This may explain our previously reported data that API treatment of OPC mice induced necrosis of pancreatic tumor cells validated by H&E staining [12]. The inhibition of PC cell growth by API may be due to the downregulation of PI3K/AKT and MAPK/MEK/ERK pathways as these kinases are down stream of Kras oncogene which is mutated in 90% of patients with PC [79,80]. It is interesting to note that both API and miR-155 have common apoptotic targets in cancer cells such as caspase 3, caspase 9, FAS and Bcl2 [51,81,82]. API depletes anti-apoptotic Bcl2 and increases pro-apoptotic caspase 3 whereas miR-155 increases anti-apoptotic Bcl2 and decreases pro-apoptotic caspase 3 in cancer cells [51,82]. In addition, API may inhibit topoisomerase I-catalyzed DNA re-ligation and enhance gap junctional intercellular communication through induction of phosphorylation of the ataxia-telangiectasia mutated (ATM) kinase and histone H2AX, two key regulators of the DNA damage response [83]. MiR-155, a prognostic blood-based biomarker linked to poor survival in PC patients, is known to be implicated in the development of pancreatic tumors from pancreatic intraepithelial neoplasia (PanIN) lesions to adenocarcinoma and is overexpressed in PC [23,24,84,85]. Overexpression of miR-155 promotes gemcitabine chemoresistance, while API has been shown to reduce this chemoresistance in human PC cell lines [50,86]. Our data further demonstrate that depletion of miR-155 using a miR-155 inhibitor reduced the viability of PC cells, supporting the role of miR-155 in PC growth and development. Moreover, combined API and miR-155 inhibitor treatment results in additional depletion of miR-155 and this corresponds with synergistic reduction of PC cell viability suggesting the use of both API and miR-155 inhibitor as an adjuvant therapy for PC. We are currently performing in vitro studies regarding API regulation of apoptotic pathways via targeting miR-155 in human and mouse pancreatic adenocarcinoma cells.

Our previous in vivo studies have shown that apigenin treatment in experimental models of PC induces anti-tumor immune responses by depleting proinflammatory TDFs, MDSC and protumor M2-TAM as well as upregulating tumoricidal M1 TAM and SHIP-1 expression in the tumors from mice [12,55]. Previous studies have shown that tumoricidal macrophage activity via production of iNOS/NO and CD40–CD40L interactions between macrophages and T cells, respectively, as well as induction of T-cell responses have tumoricidal effects [87]. Our results show that M1 TAM in KC-HPC-API mice exhibit upregulation of CD40, which corresponded with an increase in their production of iNOS and coincided with tumor regression. In addition, CD40 expressing macrophages interacting with CD40L on CD8^+^ T cells has been shown to enhance cytotoxic activity [87,88,89]. Our previous results show that API increases IFN-γ production from intratumoral CD8^+^ T cells as well as perforin and granzyme B and that this contributed to low tumor burden in PC-bearing mice [12]. These are the proposed mechanisms responsible for the tumor reduction induced by API in models with PC (Figure 11).

Overexpression of miR-155 has also been reported in both G-MDSC and M-MDSC derived from mice with lung cancer [90]. Dexamethasone has been shown to promote the expansion of MDSC and upregulate miR-155 expression in BM cells, with its effects abolished by depleting miR-155 [90]. In this present study, we are the first to show that apigenin treatment targets and suppresses miR-155 gene expression in the BM and pancreatic tumor, thereby elevating the SHIP-1 gene and protein expressions that corresponded with an increase in tumoricidal M1 TAM percentages and the infiltration of effector cytotoxic CD8+ T cells in tumors from mice (Figure 11). This correlated with a reduction in proportions of pro-tumor M2-TAM, G-MDSC and M2 macrophages in the BM, thereby increasing anti-tumor immune responses in mouse models of PC. It has been proposed that G-MDSC may develop into tumor associated neutrophils (TAN) in the TME [91]. These TAN can be tumoricidal N1 or protumor N2 similar to the M1 and M2 TAM nomenclature [91]. More recently, N1 TAN have been identified as a new target for the treatment of PC [91,92]. We are currently investigating G-MDSC development into TAN in the TME of pre-clinical PC models.

Downregulation of SHIP-1 expression is implicated in chronic myeloid leukemia, Crohn’s Disease, T cell leukemia, SLE, ulcerative colitis and Systemic Lupus Erythematous, both in humans and mice [26,31], as well as in preclinical models of PC [11,12,35]. SHIP-1 expression is regulated in immune and myeloid cells, including macrophages, by cytokine and chemokine signaling [30,33] and is one of the targets of miR-155 [28,30], impacting tumor immunity. It is also implicated in the regulation of macrophage polarization in SHIP-1 knockout mice, where they exhibit an immunosuppressive M2 macrophage (protumor) phenotype [93]. We have also reported the expansion of immunosuppressive M2 macrophage in SHIP-1-deficient mice [29]. Therefore, SHIP-1 acts as a tumor suppressor, preventing metastasis in pre-clinical cancer models [94]. In the present study, we are the first to show greater miR-155 expression in the BM and tumors of SHIP^KO^-HPC mice compared with SHIP^WT^-HPC mice indicating the role of SHIP-1 as tumor suppressor and miR-155 as an oncogene. This is interesting because it suggests that SHIP-1 may be acting in a negative feedback loop to repress miR-155 expression. We have reported that SHIP^KO^-HPC mice have a significant increase in M2-like TAM and a significant decrease in M1-like TAM in the tumor compared with SHIP^WT^-HPC mice [12]. Our results support the notion that enhanced SHIP-1 expression reduces immunosuppressive TAM and that this promotes anti-tumor immune responses in the pancreatic TME. Therefore, amplification of SHIP-1 expression through suppression of miR-155 by API or miR-155 inhibitor or both may be a novel means to enhance the anti-tumor immune responses in the pancreatic TME. We are currently performing studies to identify other SHIP-1 regulators such as miR-210 and post-translational modification events such as proteasome degradation [26,95,96,97] that could skew M-MDSC and the balance of M1 vs. M2 TAM in the tumors from PC mice.

miR-155 regulates a plethora of biological properties which include Toll-like receptor (TLR) activation on monocytes and macrophages that facilitate pro-inflammatory cellular responses [98]. We are currently exploring changes in TLR expression on TAM subsets from our API and vehicle-treated PC models. In addition, miR-155 directly targets and transcriptionally suppresses Suppressor of Cytokine Signaling 1 (SOCS1) that influence the development of immunosuppressive regulatory T cells (Treg) and Th17 cells [99]. We reported a significant reduction in intratumoral and splenic Treg percentages from our PC mouse model treated with API [12]. Wang et al., have shown that the overexpression of miR-155 coincides with an increase in Th17/Treg percentages in the blood of patients with acute pancreatitis [99]. Huang et al., reported that miR-155 enhanced pancreatic cancer cell invasiveness by modulating the STAT3 signaling pathway through SOCS1 [100]. Therefore, miR-155 regulation of SHIP-1 and SOCS1 expressions may be potential biomarkers and therapeutic targets for the treatment of pancreas-associated diseases (i.e., pancreatitis and pancreatic cancer).

We are aware of the following minor limitations related to the lack of data regarding miR-155 directly targeting and suppressing SHIP-1 transcription using murine in vitro and in vivo systems. However, it has already been reported that miR-155 targets and binds to the 3′-UTR regions of SHIP-1 transcript (which is a highly conserved binding site), in vitro, via luciferase reporter assay [28,101]. In addition, O’Connell R.M. et al. have reported that retroviral expression of miR-155, in vivo, targets SHIP-1 which alters the hematopoietic compartment causing a phenotype similar to myeloproliferative disorder (MPD) [28]. These authors have also demonstrated a similar MPD phenotype when silencing SHIP-1 using siRNA against SHIP-1, in vivo [28]. We have also reported similar findings of MPD in transgenic SHIP-KO mice and mouse models of PC [11,12,29]. In addition, we have reported that miR-155 expressing PC cells co-cultured with control splenocytes suppressed SHIP-1 gene and protein expression, in vitro [11]. Thus, the addition of miR-155 binding to the 3′-UTR regions of SHIP-1 mRNA via luciferase reporter assay data to the present study is unlikely to add any novelty to what is currently reported in the literature. However, we are planning to perform these experiments using pharmacologic and genetic tools to mechanistically show that PC-induced miR-155 targets and downregulates SHIP-1 expression using in vitro and in vivo model system.

The association between dietary flavonoids, including apigenin, and their role in cancer was investigated in patients with ovarian cancer [102], breast cancer [103], and lung cancer [104]. The data show an inverse association between the intake of flavonoids, including apigenin, and incidence of many types of solid cancers. In a prospective cohort study of patients with resected colon cancer, those treated with a flavonoid mixture (apigenin and EGCG) had a reduced rate of recurrence of colon neoplasia [105]. One open-label clinical study tested a combination therapy including apigenin, ferulic acid, gamma oryzanol, and silymarin in patients with Alzheimer’s, Parkinson’s and multiple sclerosis [106]. The data show an improvement in the pathology neurodegenerative disease. An ongoing clinical trial has been testing an apigenin-rich celery-banana bread in high-risk breast cancer patients (ClinicalTrials.gov Identifier NCT03139227). A phase I clinical trial of LNA-modified anti-miR-155 (MRG-106) has also been initiated (ClinicalTrials.gov Identifier NCT02580552). This trial has been evaluating patients diagnosed with cutaneous T-cell lymphoma (CTCL) of the mycosis fungoides subtype, and the results are encouraging. Preliminary data demonstrate that intra-tumoral injection of MRG-106 results in improved cutaneous lesions with almost no side effects. In 2018, a phase II clinical trial was also initiated to further evaluate the efficacy of MRG-106 against CTCL (ClinicalTrials.gov Identifier NCT03713320). Our preclinical data using apigenin and miR-155 inhibitor as antineoplastic agents strongly suggest the clinical utility of these agents as adjuvants in PC patients.

## 5. Conclusions

Our pre-clinical results indicate that modulating miR-155 in PC to augment SHIP-1 expression may be a mechanism to enhance anti-tumor immunity and improve immunotherapy responses for PC. It is important to note that SHIP-1 directly and indirectly regulates multiple MDSC-TAM associated signaling events that can impact tumor outcomes and warrants further investigation. More importantly, the restoration of SHIP-1 expression is an ideal therapeutic strategy to regulate myelopoiesis and promote the development of activated tumoricidal M1 TAM for the triggering of effector CD8^+^ T cells to kill PC cells (Figure 11). Thus, targeting miR-155/SHIP-1 may be a promising new approach for adjuvant therapy for the treatment of pancreatic cancer, and its activity in other solid tumors warrants further exploration.

## Figures and Tables

**Figure 1 cancers-14-03613-f001:**
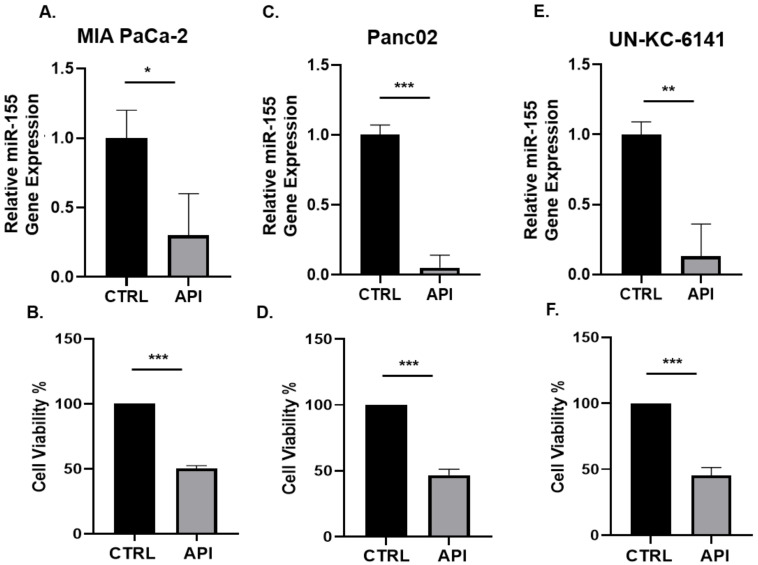
API suppressed miR-155 gene expression and inhibited cell viability in PC cells in vitro. The indicated human and mouse PC cell lines were treated with API (40 µM) or vehicle (CTRL) for 24 h and assessed for (**A**,**C**,**E**) miR-155 gene expression by qPCR or (**B**,**D**,**F**) cell viability by MTT assay. Data presented as the mean ± S.D. of each experimental group (*n* = 3). * *p* < 0.05; ** *p* < 0.001; *** *p* < 0.001 (by two-tailed *t* test).

**Figure 2 cancers-14-03613-f002:**
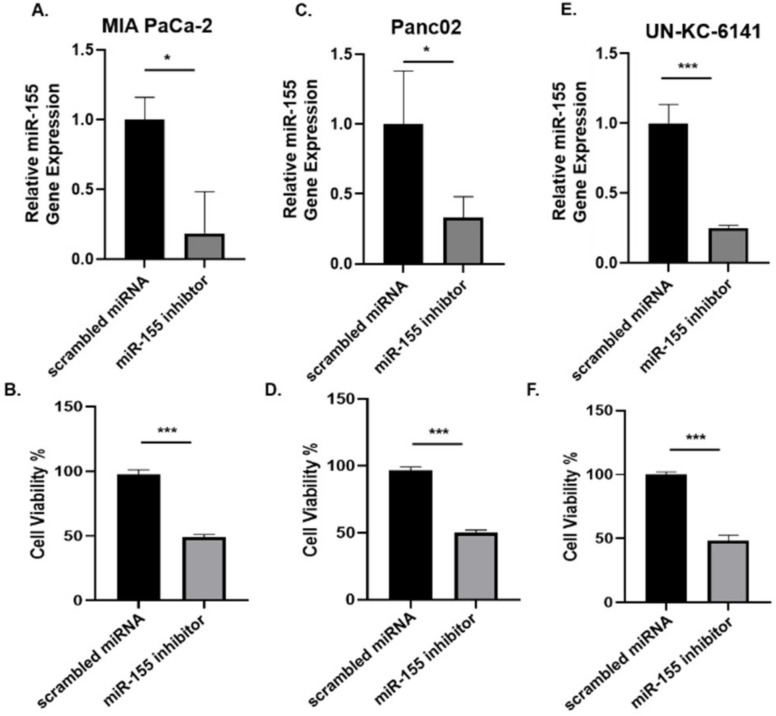
Inhibition of miR-155 expression decreased the viability of PC cells in vitro. The indicated human and mouse PC cells were treated with miR-155 inhibitor or scrambled miRNA (100 nM) for 24 h and assessed for (**A**,**C**,**E**) miR-155 gene expression by qPCR or (**B**,**D**,**F**) cell viability by MTT assay. Data presented as the mean ± S.D. of each experimental group (*n* = 3). * *p* < 0.05; *** *p* < 0.001 (by two-tailed *t* test).

**Figure 3 cancers-14-03613-f003:**
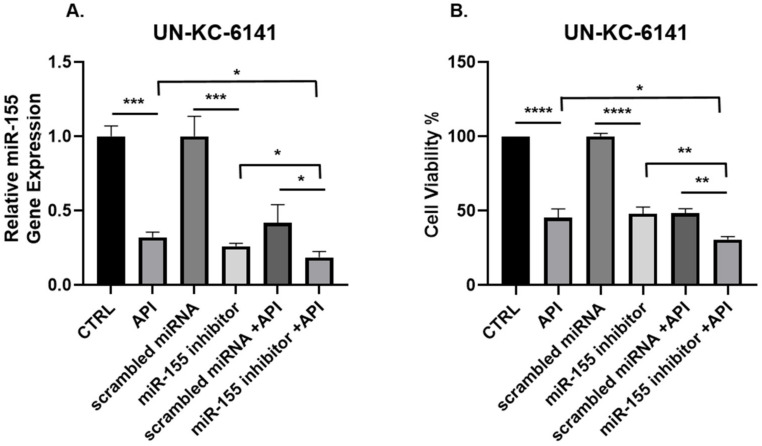
miR-155 gene expression and cell viability are synergistically suppressed in murine UN-KC-6141 cells after combined treatment with API and miR-155 inhibitor. UN-KC-6141 cells were treated with miR-155 inhibitor (100 nM), API (40 µM), their respective controls or in combination and assessed for (**A**) miR-155 gene expression or (**B**) cell viability. Data are presented as the mean ± S.D. of each experimental group (*n* = 3). * *p* < 0.05; ** *p* < 0.001; *** *p* < 0.001; **** *p* < 0.0001 (by two-tailed *t* test).

**Figure 4 cancers-14-03613-f004:**
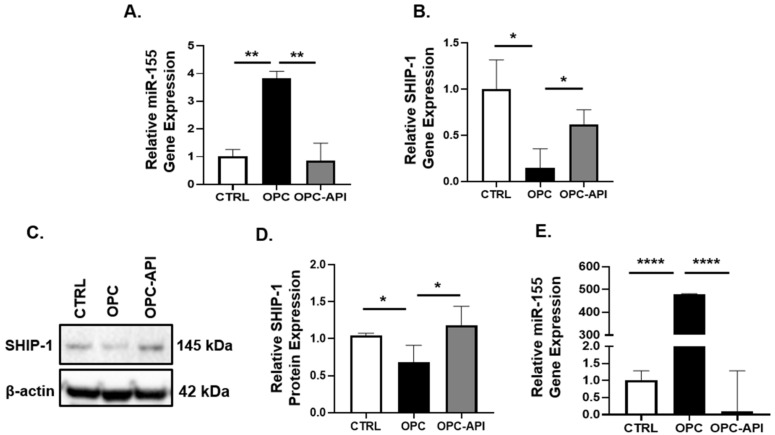
API treatment decreased miR-155 expression in OPC mice, which correlated with increased SHIP-1 expression. Relative (**A**) miR-155 and (**B**) SHIP-1 gene expression in the BM of CTRL, OPC and OPC-API mice. (**C**,**D**) WB analysis and representative quantification of SHIP-1 protein in the BM of CTRL, OPC and OPC-API mice. (**E**) Relative miR-155 gene expression in the pancreas or tumor of CTRL, OPC and OPC-API mice. Data are presented as the mean ± S.D. of CTRL (*n* = 3), OPC (*n* = 3–4), OPC-API (*n* = 3–4) mice. * *p* < 0.05; ** *p* < 0.001; **** *p* < 0.0001 (by two-tailed *t* test).

**Figure 5 cancers-14-03613-f005:**
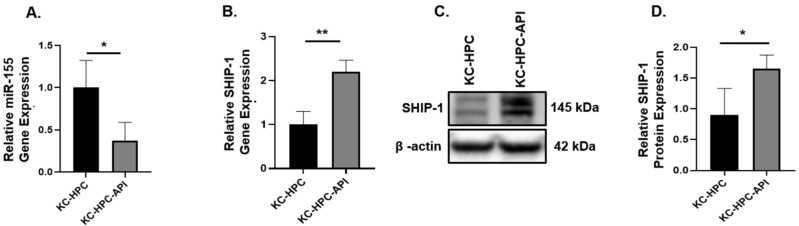
API treatment decreased miR-155 expression in KC-HPC mice, which correlated with an increase in SHIP-1 expression. Relative (**A**) miR-155 and (**B**) SHIP-1 gene expression in the tumors from KC-HPC and KC-HPC-API mice. (**C**,**D**) WB analysis and representative quantification of SHIP-1 protein from the tumors of KC-HPC and KC-HPC-API mice. Data are presented as the mean ± S.D. of KC-HPC (*n* = 3–4), KC-HPC-API (*n* = 3–4). * *p* < 0.05; ** *p* < 0.01 (by two-tailed *t* test).

**Figure 6 cancers-14-03613-f006:**
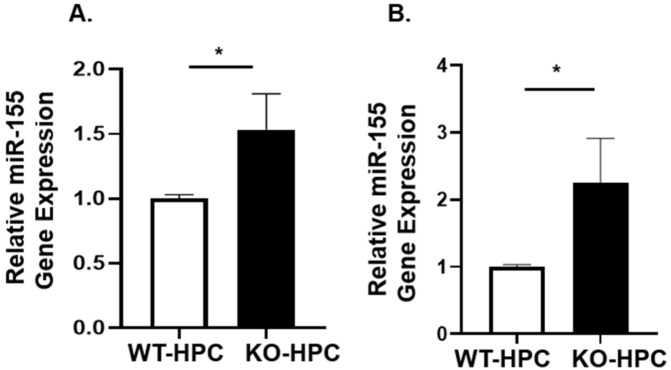
miR-155 gene expression is increased in the BM and Tumor of SHIP^KO^-HPC mice. Relative quantification of miR-155 gene expression in the (**A**) BM and (**B**) tumors from SHIP^WT^-HPC and SHIP^KO^-HPC mice. Data are presented as the mean ± S.D. SHIP^KO^-HPC (*n* = 3) and SHIP^WT^-HPC (*n* = 3) mice. * *p* < 0.05 (by two-tailed *t* test).

**Figure 7 cancers-14-03613-f007:**
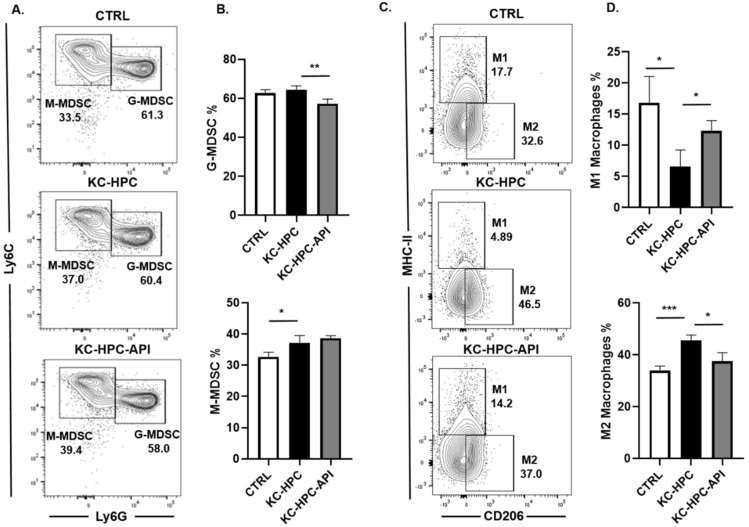
API treatment of KC-HPC mice modulated MDSC and macrophage subsets in the BM. Flow cytometric analysis and representative quantification of (**A**,**B**) MDSC subsets, G-MDSC (CD11b^+^Ly6C^+/−^Ly6G^+^) and M-MDSC (CD11b^+^Ly6G^−^Ly6C^+^), and (**C**,**D**) macrophage subsets, M1 (CD11b^+^Ly6C^+/−^Ly6G^−^F4/80^+^CD206^−^MHCII^+^) and M2 (CD11b^+^Ly6C^+/−^Ly6G^−^F4/80^+^CD206^+^MHCII^−^), from the BM of CTRL, KC-HPC and KC-HPC-API treated mice. Data are presented as the mean ± S.D. of CTRL (*n* = 3–4), KC-HPC (*n* = 4) and KC-HPC-API (*n* = 3–4) mice. * *p* < 0.05; ** *p* < 0.01; *** *p* < 0.001 (by two-tailed *t* test).

**Figure 8 cancers-14-03613-f008:**
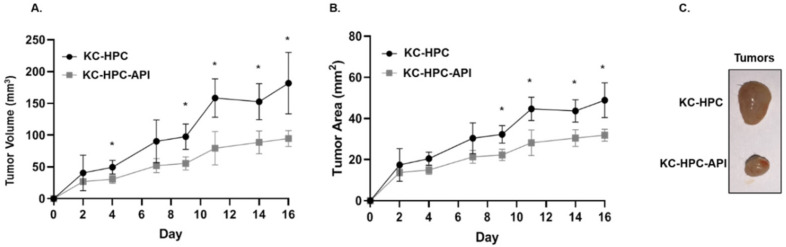
API treatment of KC-HPC mice decreased tumor burden. Tumor growth curves representing (**A**) tumor volume and (**B**) tumor area in KC-HPC mice treated with API. (**C**) Representative images of KC-HPC and KC-HPC-API tumors at the humane endpoint of the study. Data are presented as the mean ± S.D. of KC-HPC (*n* = 3–4) and KC-HPC-API (*n* = 3–4) mice. * *p* < 0.05 (by two-tailed *t* test).

**Figure 9 cancers-14-03613-f009:**
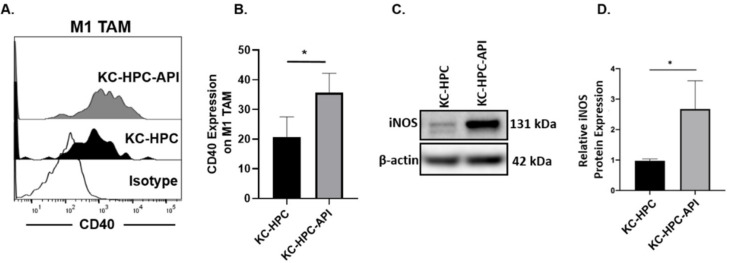
API treatment of KC-HPC mice increased CD40 expression on M1 TAM and iNOS in the tumor. (**A**,**B**). Flow cytometric analysis of CD40 expression on M1 TAM (CD11b^+^Ly6C^+/−^Ly6G^−^F4/80^+^CD206^−^MHCII^+^) in the tumors of KC-HPC and KC-HPC-API mice. (**C**,**D**) WB analysis and representative quantification of iNOS protein in the tumors from KC-HPC and KC-HPC-API mice. Data are presented as the mean ± S.D. of KC-HPC (*n* = 4), KC-HPC-API (*n* = 4) mice. * *p* < 0.05 (by two-tailed *t* test).

**Figure 10 cancers-14-03613-f010:**
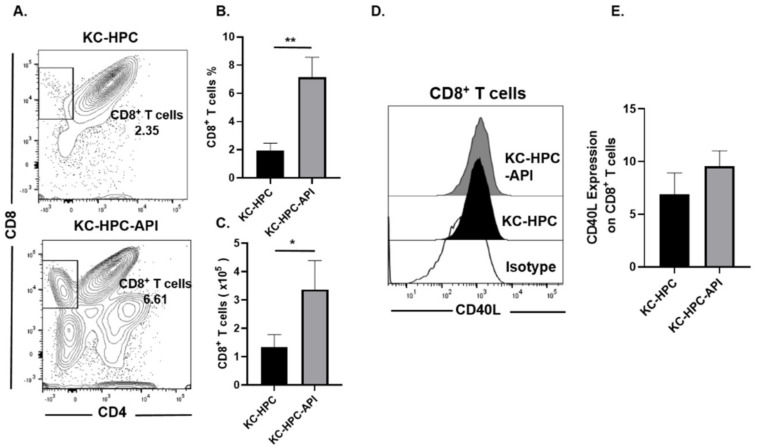
API treatment of KC-HPC mice increased CD8^+^ T cell infiltration into the tumor. (**A**,**B**) Flow cytometric analysis and (**C**) absolute cell number of CD8^+^ T cells (CD3^+^CD8^+^CD4^−^) and its expression of (**D**,**E**) CD40L in the tumors of vehicle-treated KC-HPC mice and API treated KC-HPC mice. Data are presented as the mean ± S.D. of KC-HPC (*n* = 4), KC-HPC-API (*n* = 4) mice. * *p* < 0.05; ** *p* < 0.01 (by two-tailed *t* test).

**Figure 11 cancers-14-03613-f011:**
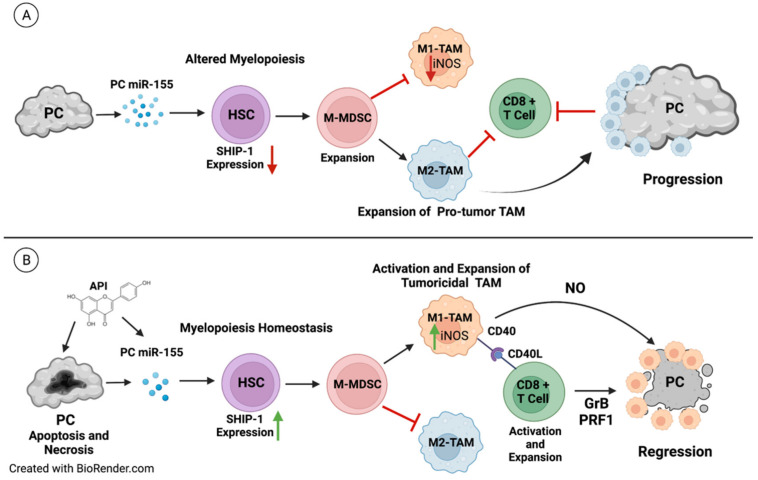
Proposed Model: (**A**) In the PC tumor microenvironment, tumor derived factors such as PC-induced miRNA-155 impacts hematopoietic stem cells (HSC) and alters myelopoiesis by targeting and downregulating SHIP-1 gene and protein expressions. The reduction in SHIP-1 expression corresponds with the expansion of M-MDSC that correspond with the development of pro-tumor M2-TAM which, in turn, impair anti-tumor immunity and result in PC progression and metastasis. (**B**) The therapeutic use of the bioflavonoid API induces apoptosis and necrosis of PC cells and causes a decrease in miRNA-155 in the pancreatic tumor. This reduction correlates with an increase in SHIP-1 gene and protein expressions that results in the expansion of M-MDSC which coincides with the development of tumoricidal M1-TAM. These tumoricidal M1-TAM have increased CD40 expression which interacts with CD40L on CD8^+^ T cells, resulting in M1-TAM activation, production of iNOS/NO and an overall increase in anti-tumor immunity. This is signified by the release of granzyme B (GrB) and perforin (PRF) as well as the robust activation of effector CD8^+^ T cells, leading to PC regression. Created with BioRender.com (accessed on 21 March 2022).

**Table 1 cancers-14-03613-t001:** miRNA-155 inhibitors.

Qiagen miRCURY LNA miRNA Inhibitors	Sequence 5′–3′
MMU-MIR-155-5P (mouse)	CCCCTATCACAATTAGCATT
HSA-MIR-155-5P (human)	CCTATCACGATTAGCATT

**Table 2 cancers-14-03613-t002:** Lower survival probability in PC patients assessed at a single time point for high miR-155HG and low SHIP-1 expression (Appendix A).

Survival Analysis Type	Time Period	Two Proportion Test *p*-Value
Overall survival	10 months	0.006
Disease specific	10 months	0.078

## Data Availability

Data are contained within the article and Appendix A.

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
