# Peer review of "Apigenin Targets MicroRNA-155, Enhances SHIP-1 Expression, and Augments Anti-Tumor Responses in Pancreatic Cancer"

_cancers, 2022, doi:10.3390/cancers14153613_

Round 1
Reviewer 1 Report
- A brief summary
Based on the previous findings from this research group that apigenin increases SHIP-1 expression, promotes tumoricidal macrophages and anti-tumor immune responses in murine pancreatic cancer, this is a follow-up study that tries to elucidate the mechanism by which apigenin regulates SHIP-1 expression and augments the anti-tumor response. Using both human and mouse pancreatic cancer cell lines, the authors found that API could suppress miR-155 expression and inhibit cell viability in PC cells in vitro. By further employing the heterotopic and orthotopic mouse models, the authors found that API treatment decreases miR-155 expression in the TME or BM of these mouse models, which also correlates with an increase in the SHIP-1 expression. In addition, API treatment leads to a decreased MDSC cell count and an increased M1 macrophage percentage in the BM as well as increased CD40 expression on M1 macrophages and increased CD8+ T cell percentage in the TME of the heterotopic mouse model, all of which contribute to the decreased tumor burden. This study might provide some insights into the prognostic and therapeutic applications, but further improvements could be done before publication.
- General concept comments
First of all, it has been shown in this manuscript that API suppresses miR-155 expression and induces apoptosis in the cell lines. What are the target genes of miR-155 that contribute to cell apoptosis in these cell lines ? Also, does miR-155 overexpression in these cells abrogate the API-induced cell apoptosis ?
Secondly, while using these different mouse models, it’s important to examine both the TME and BM (where HSC mainly reside in). For example, how about the miR-155 / SHIP-1 expressions in the BM and the MDSC / macrophage subsets in the TME of KC-HPC mouse model ? In addition, according to the proposed model, miR-155 secreted by cancer cells impacts HSC in the BM. That said, is there elevated miR-155 detected in the blood of diseased mouse models compared to control ?
In addition, in Fig 9, it says that “ API treatment of KC-HPC mice increased iNOS and CD40 expression on M1 TAM in the TME ”. However, iNOS protein expression was examined using the whole tumor tissue (TME) rather than M1 macrophages. Although iNOS is a marker of M1 macrophage activation, it could also be expressed by many other types of immune cells and nonimmune cells. If the purpose here is to examine iNOS as a M1 macrophage activation marker, then M1 macrophages could be intracellularly stained with an iNOS antibody for flow cytometry analysis.
Finally, have other immune populations such as NK cells and Treg ever been examined in this study ? For example, miR-155 has been shown to play a role in tuning the threshold and extent of NK cell activation, and both miR-155 and SHIP have been found to regulate Treg development. Also, how about cytokine expressions such as TGF-a and GM-CSF in the TME and BM of mouse models ?
- Specific comments
- Are there higher miR-155 expressions in the pancreatic cancer cell lines compared to a normal pancreatic cell line ?
- In Fig S1, there is no significant difference in the cell apoptosis between CTRL and API 50uM groups for the UN-KC-6141 cell line, what are the possible explanations ?
- In Fig 3, is there any significant difference when comparing miR-155 inhibitor+API to miR-155 inhibitor alone or API alone ?
- In Fig 4, if there is a CTRL+API group, is this group expected to have lower miR-155 and higher SHIP-1 compared to CTRL group ?
- In Fig 5 legend, it says “ gene expression in the TME of CTRL…” , but there is no CTRL group shown in the panels.
- What are the expected results if SHIP-HPC mice are treated with API ? Also, in the SHIPKO-HPC mice where miR-155 expression increased compared to the WT, are the gene expressions of other miR-155 targets expected to be downregulated ?
- In Fig 10, I assume the CD8+ T cell population is gated out of total CD3+ T cells. If so, this is just showing the CD8+ T cell percentage out of total T cells. How about the absolute number of CD8+ T cells that infiltrated into the TME ?
Author Response
First of all, it has been shown in this manuscript that API suppresses miR-155 expression and induces apoptosis in the cell lines. What are the target genes of miR-155 that contribute to cell apoptosis in these cell lines? Also, does miR-155 overexpression in these cells abrogate the API-induced cell apoptosis?
miR-155 has several specific apoptotic target genes (i.e., Caspase 3, Caspase-9, BID, FoxO3a, BIM, Bcl-2, TP53INP1, FAS, APAF1, and PTEN, etc.) that confers resistance of apoptosis of plethora cancer cell lines (https://doi.org/10.3390/genes10100752, doi: 10.3390/molecules24224153, doi: 10.12701/yujm.2020.00836). We currently have a separate manuscript that focuses of API regulation of apoptotic pathways via targeting miR-155 in human and mouse pancreatic adenocarcinoma cells (Krystal Villalobos and Ana Olavaria et al 2022). In this revised manuscript, we have added supplemental material which shows that API significantly reduced Bcl-2 expression in UN-KC-6141 cells (relevant humans with PC due to Kras mutation) (Figure S1).
In some cases, over expression of miR-155 in hematological (blood) cancers induces apoptosis (DOI: 10.1186/1476-4598-13-79). Over-expression of Bcl-2 (anti-apoptotic) in cancer cells refers resistance to apoptosis (Campbell, KJ, et al. doi: 10.1098/rsob.180002). Our result suggest that miR-155 overexpression leads to PC cell resistance to apoptotic events due to the activation of Bcl-2 anti-apoptotic pathways (intrinsic) in which treatment with API increases their susceptibility to apoptosis. In our studies, pancreatic cancer cell lines overexpress miR-155 and BCl2 therefore this suggest that API may potentially target multiple apoptotic pathways that warrants further investigation.
Secondly, while using these different mouse models, it’s important to examine both the TME and BM (where HSC mainly reside in). For example, how about the miR-155 / SHIP-1 expressions in the BM and the MDSC / macrophage subsets in the TME of KC-HPC mouse model? In addition, according to the proposed model, miR-155 secreted by cancer cells impacts HSC in the BM. That said, is there elevated miR-155 detected in the blood of diseased mouse models compared to control?
Thank you for the keen observation. We have not performed these studies in the BM of KC mice. However, we have reported in this manuscript (see Figure 4) that miR-155/SHIP-1 is expressed in BM from orthotopic PC model. In addition, we reported the frequency of MDSC/Macrophage subsets in tumor of KC-HPC (Cancers 2020). Currently, we have not performed specific isolation of exosomes for detection miR-155 from blood serum of our PC mice. However, Yasbushita et al reported that serum miR-155 was higher in PDAC rats, similar to plasma of patients with PDAC compared to normal rats (Yabushita S. et al. 2012. DOI: 10.1097/MPA.0b013e31824ac3a5).
In addition, in Fig 9, it says that “API treatment of KC-HPC mice increased iNOS and CD40 expression on M1 TAM in the TME”. However, iNOS protein expression was examined using the whole tumor tissue (TME) rather than M1 macrophages. Although iNOS is a marker of M1 macrophage activation, it could also be expressed by many other types of immune cells and nonimmune cells. If the purpose here is to examine iNOS as a M1 macrophage activation marker, then M1 macrophages could be intracellularly stained with an iNOS antibody for flow cytometry analysis.
Thank you for this observation. In this experiment for Fig. 9, we did not intracellularly stain for M1 TAM with iNOS antibody. Therefore, we have corrected the figure legend in this revised manuscript to reflect that accordingly.
Finally, have other immune populations such as NK cells and Treg ever been examined in this study? For example, miR-155 has been shown to play a role in tuning the threshold and extent of NK cell activation, and both miR-155 and SHIP have been found to regulate Treg development. Also, how about cytokine expressions such as TGF-a and GM-CSF in the TME and BM of mouse models?
This is a great observation. Yes, we have immunophenotyped and detected significant increase in NK cell percentages in tumors from API treated PC mice compared non-treated PC mice. We previously reported that Treg percentages from the tumor and spleen were reduced in PC mice treated with API as compared to non-treated PC mice (Cancers 2020). We have not yet assessed TGF-alpha or GM-CSF from tumor or BM in our PC models.
- Specific comments
Are there higher miR-155 expressions in the pancreatic cancer cell lines compared to a normal pancreatic cell line?
Bloomston et al., reported that microRNA-155 expression patterns are higher in pancreatic adenocarcinoma, chronic pancreatitis as compared to normal pancreas in human (Bloomston M et al. 2007; doi: 10.1001/jama.297.17.1901). Papaconstantiou et al., reported that pancreatic cancer patient tumor tissues have an up-regulation of miR-155 compared to healthy pancreatic tissues and can be used as a potential biomarker (doi: 10.1097/MPA.0b013e3182592ba7). In addition, elevated expression of miR-155 in pancreatic tumors was associated with poorer survival in cancer patients (Greither T et al. 2010, DOI: 10.1002/ijc.24687). These PC clinical results coincided with our pre-clinical results reported in this revised manuscript showing that of miR-155 production being higher in tumors of PC mice in comparison to healthy pancreas from control mice (Please see Figure 4E).
Indeed, thanks for this observation. We have established a collaboration with Drs. Surinder Batra/Mokenge Malafa to secure HPNE (human pancreatic normal epithelial cells) or HPDE (human pancreatic ductal epithelial cells) control normal pancreatic cell line to perform the experiment in Ghansah research lab.
- In Fig S1, there is no significant difference in the cell apoptosis between CTRL and API 50uM groups for the UN-KC-6141 cell line, what are the possible explanations?
There are several explanations for no significant difference observed in UN-KC-6141 cell apoptosis treated with 50mM API. First UN-KC-6141 cells have two genetic mutations of oncogene Kras and PDX-1. These cells are more resistant to API induced apoptosis compared to Panc02 cells which harbor mutation in tumor suppressor SMAD4 (DPC4) gene. The other explanation may be due to the induction of autophagy at higher concentration of API in these cells. Apigenin 200 µM has been reported to induce autophagy rather than apoptosis in leukemia cell line (Ruela-de-Sousa, R. R. et al. 2010; doi: 10.1038/cddis.2009.18).
- In Fig 3, is there any significant difference when comparing miR-155 inhibitor+API to miR-155 inhibitor alone or API alone?
Yes, there is a significant difference when comparing miR-155 inhibitor+ API to miR-155 inhibitor or API alone. We have revised Fig. 3 showing the statistical changes between the requested groups.
- In Fig 4, if there is a CTRL+API group, is this group expected to have lower miR-155 and higher SHIP-1 compared to CTRL group?
If there was a CTRL+API group, we expected API will not lower miR-155 and will not increase the SHIP-1. In view of the fact that the CTRL mice do not have tumors and miR-155 will be extremely low, API will not be able to further decrease the miR-155 levels.
- In Fig 5 legend, it says “gene expression in the TME of CTRL…” , but there is no CTRL group shown in the panels.
Thanks for this observation. We have corrected the legend of Fig. 5 in this revised manuscript.
- What are the expected results if SHIP-HPC mice are treated with API? Also, in the SHIPKO-HPC mice where miR-155 expression increased compared to the WT, are the gene expressions of other miR-155 targets expected to be downregulated?
SHIP-HPC mice treated with API should show a decrease in miR-155 levels similar to our HPC models reported in this revised manuscript (Please see Figure S2). We expect other miR-target genes such as Suppressor of Cytokines Signaling 1 (SOCS1), Phosphatase Tensin Homology (PTEN), to be downregulated in both SHIPKO-HPC vs. WT HPC. We are currently breeding for SHIP KO and WT mice to perform the proposed experiments.
- In Fig 10, I assume the CD8+ T cell population is gated out of total CD3+ T cells. If so, this is just showing the CD8+ T cell percentage out of total T cells. How about the absolute number of CD8+ T cells that infiltrated into the TME?
Thanks for this observation. We have revised Fig. 10 and included the absolute cell numbers of intratumoral CD8+ T cells from API treated and non-treated PC mice.
Reviewer 2 Report
Dear authors,
This study is good and needs to be improved for our audience. It contains more lack of literature evidence which is not possible to complete the story here and validated methodology and need to be improved further to build this study strong enough to publish.
I have a few minor suggestions that the authors might consider, but all of them would be moving forward.
- Apigenin, which is a plant flavonoid; found to inhibit cell proliferation by arresting the cell cycle at the G2/M phase in the cell cycle and most important also activate the secondary receptor cellular pathways which is not mentioning here in this manuscript.
- Also, Apigenin directly inhibit the growth through cell cycle arrest and induction of apoptosis appear to be related to induction of p53.
- Further, Inhibition of PMA-mediated tumor promotion by inhibiting protein kinase C and the resulting suppression of oncogene expression. Can the authors discuss this point and show the gene's expression?
- It has also been reported to inhibit topoisomerase I-catalyzed DNA re-ligation and enhance gap junctional intercellular communication. Please provide the data to show these comments to better understating and missing.
Author Response
- Apigenin, which is a plant flavonoid; found to inhibit cell proliferation by arresting the cell cycle at the G2/M phase in the cell cycle and most important also activate the secondary receptor cellular pathways which is not mentioning here in this manuscript.
The following paragraph has now been added in the introduction of the revised manuscript: Recently, natural compounds known as bioflavonoids including apigenin (API) have been demonstrated both in vitro and in vivo models to exert broad anticancer activities in a variety of malignancies such as breast cancer [39], liver cancer [40], prostate cancer [41], lung cancer [42], colon cancer [43], melanoma [44], osteosarcoma [45] and PC [12,46,47]. API inhibits tumor cell proliferation by inducing apoptosis leading to autophagy and cell cycle arrest at the G2/M phase and can also reduce cancer cell motility, thereby preventing cancer cell migration and invasion regulating PI3K/AKT, MAPK/ERK, JAK/STAT, NF-κB, p53 and Wnt/β-catenin signaling pathways [48,49].
Apigenin has been shown to induce apoptosis through p53-dependent and p53-independent mechanisms (Jang, JY et al. 2022, https://doi.org/ 10.3390/ijms23073757). The induction of apoptosis induced by API involves both extrinsic and intrinsic pathways in cancer cells (Oishi, M 2013, https://doi.org/10.1371; Chen, M 2016, doi: 10.1038/srep35468). Apoptosis targets of API consists of caspase-3, -8, and -9, Bax, Bak, Bad, Bim, , Bid, Bcl-xL, XIAP, Mcl-1, , Bcl-2, m-TOR/PI3K/AKT, STAT3, p53, p21, p27, PARP cleavage, FOXO3a, AIF, Apaf-1, DR5, ERK/JNK/p38 MAPK, Jun, NF-κB, Noxa, PUMA, Smac, Survivin, FAS and TRAIL ( Jang, JY et al. 2022, https://doi.org/ 10.3390/ijms23073757).
Further, Inhibition of PMA-mediated tumor promotion by inhibiting protein kinase C and the resulting suppression of oncogene expression. Can the authors discuss this point and show the gene's expression?
Apigenin likely to inhibit PMA-mediated tumor promotion by inhibiting the protein kinase C thereby C-June/C-Fos oncogenes and kinases such as pAKT and CK2. We have reported earlier that API inhibits CK2 in pancreatic cancer (Husain et al 2020).
- It has also been reported to inhibit topoisomerase I-catalyzed DNA re-ligation and enhance gap junctional intercellular communication. Please provide the data to show these comments to better understating and missing.
Currently, we do not have the data to show the role of API in the inhibition of topoisomerase I-catalyzed DNA re-ligation in pancreatic cancer. We have address this as a potential mechanism in the discussion as follows: Apigenin inhibits topoisomerase I-catalyzed DNA re-ligation and enhance gap junctional intercellular communication through induction of phosphorylation of the ataxia-telangiectasia mutated (ATM) kinase and histone H2AX, two key regulators of the DNA damage response (https://doi.org/10.1016/j.biocel.2013.10.004) (Please see lines: 509-512).
Reviewer 3 Report
In the current manuscript, Husain et al. investigated the mechanism through which Apigenin (API) suppresses pancreatic cancer development. Specifically, they found that API inhibited miR-155 expression leading to enhanced SHIP-1 expression and increase anti-tumor immune response in the tumor microenvironment. Overall, the manuscript is well-written, the experimental design is logical, and the data presented to support the authors’ claims are convincing. However, some data do not really fit well with the model proposed by the authors (Fig. 11). Below are my comments for the present study:
- In Fig. 1, Fig. 2, and Fig. 3 where the authors either treated PC cell lines with API, knock-down miR-155 or combined treatment of both API and miR-155 knockdown, the experiments were performed in monoculture of just the pancreatic cancer cells. According to the authors’ proposed model in Fig. 11, API works through inhibiting miR-155 and enhancing the anti-tumor immune environment by directing M-MDSC toward the tumoricidal M1-TAM. However, as mentioned above, Fig. 1, 2, 3 are all with monoculture of pancreatic cancer cells alone and not co-culture with any M-MDSC or other immune cells. Yet, API or miR-155 knock-down alone or in combination was able to reduce cell viability and increase apoptosis (Fig. S1) in these pancreatic cancer cells alone. These data suggest that API or miR-155 knock-down has other mechanisms to kill pancreatic cancer cell independent of all the immune activation proposed by the authors.
- Fig. S2 should be a main Figure since it contains important in vivo data to support the authors’ main claims while Fig. 3 should be a supplemental figure since it doesn’t add much to the story.
- Throughout the study, the authors keep referring to the Bone Marrow (BM) and Tumor Microenvironment (TME) in the context of the samples analyzed. For examples: (Page 11 Line 376) Figure 6. miR-155 gene expression is increased in the BM and TME of SHIP-KO-HPC mice. I think this can cause potential misunderstanding to the readers. If I understand correctly, BM refers to the cells isolated from the bone marrow of the mouse and TME is the actual tumors that the authors removed from the mouse and analyzed. However, TME is a specific term that most readers will associate with an “environment”. Thus, using the term TME for the “tumor” could potentially mislead the readers into thinking that the authors somehow manage to isolate just the tumor microenvironment alone and analyze that. I apologize in advance if I somehow misunderstand the method or mis-interpret this “TME” term that I think the authors use to refer to the actual “tumor”.
- The procedure to isolate the bone marrow cells is not described in the Materials and Methods section.
- Please provide the sequences for the miR-155 inhibitor (MMU-MIR-155-5P/HAS-MIR-155-5P) if available.
Author Response
- In Fig. 1, Fig. 2, and Fig. 3 where the authors either treated PC cell lines with API, knock-down miR-155 or combined treatment of both API and miR-155 knockdown, the experiments were performed in monoculture of just the pancreatic cancer cells. According to the authors’ proposed model in Fig. 11, API works through inhibiting miR-155 and enhancing the anti-tumor immune environment by directing M-MDSC toward the tumoricidal M1-TAM. However, as mentioned above, Fig. 1, 2, 3 are all with monoculture of pancreatic cancer cells alone and not co-culture with any M-MDSC or other immune cells. Yet, API or miR-155 knock-down alone or in combination was able to reduce cell viability and increase apoptosis (Fig. S1) in these pancreatic cancer cells alone. These data suggest that API or miR-155 knock-down has other mechanisms to kill pancreatic cancer cell independent of all the immune activation proposed by the authors.
–Yes, API is acting on PC cells directly and impact immune myelopoiesis indirectly as stated in the proposed model (Figure 11) in this revised manuscript.
- S2 should be a main Figure since it contains important in vivo data to support the authors’ main claims while Fig. 3 should be a supplemental figure since it doesn’t add much to the story.
In this manuscript, we have added data using Panc02 cells development into an Orthotopic PC model. Therefore, it will be redundant to add Fig. S2 to the main manuscript, which is different, because it shows data using our heterotopic PC model.
- Throughout the study, the authors keep referring to the Bone Marrow (BM) and Tumor Microenvironment (TME) in the context of the samples analyzed. For examples: (Page 11 Line 376) Figure 6. miR-155 gene expression is increased in the BM and TME of SHIP-KO-HPC mice. I think this can cause potential misunderstanding to the readers. If I understand correctly, BM refers to the cells isolated from the bone marrow of the mouse and TME is the actual tumors that the authors removed from the mouse and analyzed. However, TME is a specific term that most readers will associate with an “environment”. Thus, using the term TME for the “tumor” could potentially mislead the readers into thinking that the authors somehow manage to isolate just the tumor microenvironment alone and analyze that. I apologize in advance if I somehow misunderstand the method or mis-interpret this “TME” term that I think the authors use to refer to the actual “tumor”.
Great observation. We have revised the manuscript to address the TME as pancreatic tumors or tumor throughout the document.
- The procedure to isolate the bone marrow cells is not described in the Materials and Methods section.
Great observation. We have revised material and methods regarding the procedure for the isolation of bone marrow cells (Please see lines 205-210).
- Please provide the sequences for the miR-155 inhibitor (MMU-MIR-155-5P/HAS-MIR-155-5P) if available.
Please find the sequences from miR-155 inhibitor found in the materials and methods (Please see Table 1, Page 4).
Reviewer 4 Report
In the present study, Husain et al investigate the molecular mechanism of the bioflavonoid apigenin (API) and propose miR-155 as one of its targets. According to the proposed model, the inhibition of miR155 would explain SHIP-1 upregulation upon API treatment, revealing miR155 as a new potential target for boosting anti-tumor response. Although the idea to boost the anti-tumor responses SHIP-1-mediated in the context of API treatment was already studied in their previous manuscript, the authors aim at providing more insights on apigenin molecular mechanisms. In order to accomplish this, the data presented should be improved and be more focused on the miR155 mechanism, providing more specificity of the results observed (add-back of miR-155 in vitro and a miR-155 KO in vivo to compare with the API treated mice). Specific comments:-Fig.1 It would be useful to assess the gene expression level of miR-155 targets involved in the inflammation process (such as SOCS-1 and BCL6) in addition to SHIP-1.-Is miR-155 proven to target SHIP-1 in PCs? A reporter system (with a 3'UTR-SHIP-1 and a 3'UTR-SHIP-1 mutant) should be used to verify that SHIP1 is a real target of miR-155 in the cell lines under study.
-Fig.2 A miR-155 inhibitor is tested in this figure, but the reference (see Materials) is not correct (please give appropriate product references). What kind of inhibitor is used? -Fig.4B is redundant. The conclusions presented in the graph are the same presented in Fig. 2D of the manuscript (doi: 10.3390/cancers12123631). It should be removed from the main Figure and moved in the Supplementary. -Fig.5 Provide a higher quality WB.
-Fig.6 It will be more informative to KO miR-155 in vivo and describe the phenotype obtained in terms of SHIP-1 gene levels, CKs production, tumor volumes and M1 and M2 population. This set of experiments would help to define the role of miR-155 relative to SHIP-1 and eventually to claim specificity of the data observed. -Fig.8 Provide histology (H&E staining). -Fig.9 I would suggest to evaluate more than one marker of M1 activation.
-It would be better to substitute the data in Table1 with those of Supplementary Fig.4
Author Response
In the present study, Husain et al investigate the molecular mechanism of the bioflavonoid apigenin (API) and propose miR-155 as one of its targets. According to the proposed model, the inhibition of miR155 would explain SHIP-1 upregulation upon API treatment, revealing miR155 as a new potential target for boosting anti-tumor response. Although the idea to boost the anti-tumor responses SHIP-1-mediated in the context of API treatment was already studied in their previous manuscript, the authors aim at providing more insights on apigenin molecular mechanisms. In order to accomplish this, the data presented should be improved and be more focused on the miR155 mechanism, providing more specificity of the results observed (add-back of miR-155 in vitro and a miR-155 KO in vivo to compare with the API treated mice).
We reported in 2011 in PLOS One that the addition of Pan02 supertants (that contains miR-155) co-cultured with mouse splenocytes showed a signifinat reduction in SHIP-1 gene and protein expressions (Nelson et al 2011). These infer that PC-miR-155 transcriptionally downregulated SHIP-1 expression in vitro.
We currently have collaboration with Dr. Brad McGwire (Ohio State University) to receive breeding pair of transgenic miR-155 when they become available. We will perform these in vivo experiments with miR-155 KO and WT mice treated with and without API in the future.
Specific comments:
-Fig. 1 It would be useful to assess the gene expression level of miR-155 targets involved in the inflammation process (such as SOCS-1 and BCL6) in addition to SHIP-1.-Is miR-155 proven to target SHIP-1 in PCs? A reporter system (with a 3'UTR-SHIP-1 and a 3'UTR-SHIP-1 mutant) should be used to verify that SHIP1 is a real target of miR-155 in the cell lines under study.
Thanks for the suggestion. We currently have a separate study that focuses of API regulation of SOCS1, Bcl-6, etc. in human and mouse pancreatic adenocarcinoma cells (Krystal Villalobos and Valentina Laverde et al 2022). Also, SHIP-1 is solely expressed in hematopoietic stem cells and is not expressed in PC cell lines (doi: 10.1371/journal.pone.0027729). Currently, we are unable to perform these molecular experiments at this time but we plan to do so in the future with the purchase of SHIP-1 plasmids.
-Fig. 2 A miR-155 inhibitor is tested in this figure, but the reference (see Materials) is not correct (please give appropriate product references). What kind of inhibitor is used?
Please find that miRCURY LNA miRNA inhibitors from Qiagen were used, the methods and materials have been updated along with addition of requested sequence (Please Table 1, Page 4).
-Fig. 4B is redundant. The conclusions presented in the graph are the same presented in Fig. 2D of the manuscript (DOI: 10.3390/Cancers12123631). It should be removed from the main Figure and moved in the Supplementary.
Thank you for the observation. Figure 4b represents SHIP-1 expression in the BM while Fig. 2D of the manuscript from 2020 (doi: 10.3390/cancers12123631) represents SHIP-1 expression in the spleen.
-Fig. 5 Provide a higher quality WB.
Thanks for your observation. We have revised Fig. 5 to a higher quality image.
-Fig. 6 It will be more informative to KO miR-155 in vivo and describe the phenotype obtained in terms of SHIP-1 gene levels, CKs production, tumor volumes and M1 and M2 population. This set of experiments would help to define the role of miR-155 relative to SHIP-1 and eventually to claim specificity of the data observed.
We currently have collaboration with Dr. McGwire to share a breeding pair of miR-155 KO mice when they become available to perform the experiments suggested.
-Fig. 8 Provide histology (H&E staining).
All tumor tissues were used for biological assays and thus we did not have tumor for H & E staining. However, we have published the H&E staining of tumors from OPC mice (Villalobos-Ayala et al Cancers 2020). We plan to use KC-OPC mice in future for H&E staining of the tumor.
-Fig. 9 I would suggest to evaluate more than one marker of M1 activation.
We have provided at least two- three M1 distinguished markers MHC-II, CD40 and iNOS. In this study M1 TAM were gated from overexpression of MHC-II along with CD40 and stained for iNOS expression in the tumor. It has also been reported that M1 TAM can present antigen (doi:10.3390/cells9010046). In the future we will evaluate IFN-gamma, TNF-alpha and TLR4 expression in M1 TAM from tumor of PC mice.
-It would be better to substitute the data in Table1 with those of Supplementary Fig.4
Thanks for the observation. We observed a significant difference regarding an increase in the survival of PC patients with high SHIP-1 and low miR-155 expression 10 months post diagnoses. However, beyond 10 months there is no significant difference in survival of PC patients. PC is a grim disease with poor prognosis and majority patients succumb to the cancer less than 1 year from diagnosis. So, early detection of miR-155 and SHIP-1 expressions can serve as prognostic biomarkers and potential therapeutic targets for the treatment of pancreatic cancer.
Round 2
Reviewer 1 Report
Questions raised have been properly addressed and experiments suggested have been performed/ or to be performed. This manuscript could be considered for publication.
Author Response
Thank you for peer reviewing our research manuscript.
Reviewer 2 Report
Dear authors,
Overall work here, I believe the authors have done an exemplary job in preparing this manuscript.
Author Response

(The authors gave the same response as above.)

Reviewer 3 Report
The authors have adequately addressed all my concerns. The manuscript can be accepted for publication in its current form.
Author Response

(The authors gave the same response as above.)

Reviewer 4 Report
Although the authors responded to the comments and made some changes, they failed to address the key experimental concerns regarding the data specificity.
The only direct method to find a miRNA target is to use a reporter construct (3'UTR-luciferase) as suggested in the revision. All the other experiments are a measure of downstream effects of miRNAs which doesn't necessary prove cause-effect between the two events. It can be affected by off-targets.
Since it seems that the authors are going to address these technical concerns in the future, the publication of the present manuscript is premature. I would suggest first to publish the missing experiments (in vivo and in vitro) in the other publications, and then resubmit the present work.
Author Response
Dear Reviewer 4,
Please find on lines 643-659 our response to your comments regarding using a 3'UTR luciferase reporter system to validate SHIP-1 as a target of miR-155. In this revised manuscript discussion section to address this limitation-“We are aware of the following minor limitation of not providing data regarding miR-155 directly targeting and suppressing SHIP-1 transcription using murine in vitro and in vivo systems. However, it has already been reported that miR-155 targets and binds to the 3’-UTR regions of SHIP-1 transcript (which is a highly conserved binding site), in vitro, via luciferase reporter assay [28,102]. In addition, O’Connell R.M. et al. reported that retroviral expression of miR-155, in vivo, targets SHIP-1 which alters the hematopoietic compartment causing a phenotype similar to myeloproliferative disorder (MPD) [28]. These authors also demonstrated a similar MPD phenotype when silencing SHIP-1 using siRNA against SHIP-1, in vivo [28]. We also reported similar findings of MPD in transgenic SHIP-KO mice and mouse models of PC [11,12,29]. In addition, we reported that miR-155 expressing PC cells co-cultured with control splenocytes suppressed SHIP-1 gene and protein expression, in vitro [11]. Thus, the addition of miR-155 binding to the 3’-UTR regions of SHIP-1 mRNA via luciferase reporter assay data to the present study is unlikely to add any novelty to what is currently reported in the literature. However, we are planning to perform experiments using pharmacologic and genetic tools to mechanistically show that PC-induced miR-155 targets and downregulates SHIP-1 expression using in vitro and in vivo model systems.”
We appreciate your comments and now have addressed this limitation and provided more supporting references regarding miR-155 targeting and binding to SHIP-1 in several preclinical and clinical systems. In addition, we have provided more supporting data about MHC-II+ immune cells (confer M1 TAM) in the tumor of KC-HPC mice using state of the art technology Akoya Codex (please see Figure S4).
Round 3
Reviewer 4 Report
The authors made an effort to respond to my concerns and they provided more details about the specificity of their findings. The manuscript is now improved.